# Patatin-Related Phospholipase pPLAIIIγ Involved in Osmotic and Salt Tolerance in *Arabidopsis*

**DOI:** 10.3390/plants9050650

**Published:** 2020-05-20

**Authors:** Jianwu Li, Maoyin Li, Shuaibing Yao, Guangqin Cai, Xuemin Wang

**Affiliations:** 1College of Horticulture, Henan Agricultural University, Zhengzhou 450002, China; 2Department of Biology, University of Missouri, St. Louis, MO 63121, USA; maoyinli@hotmail.com (M.L.); sym3d@mail.umsl.edu (S.Y.); GCai@danforthcenter.org (G.C.); 3Donald Danforth Plant Science Center, St. Louis, MO 63132, USA; 4National Key Laboratory of Crop Genetic Improvement, Huazhong Agricultural University, Wuhan 430070, China

**Keywords:** patatin-related phospholipase A, lysophospholipids, free fatty acid, abiotic stress, signaling mediators, *Arabidopsis*

## Abstract

Patatin-related phospholipases (pPLAs) are acyl-hydrolyzing enzymes implicated in various processes, including lipid metabolism, signal transduction, plant growth and stress responses, but the function for many specific pPLAs in plants remains unknown. Here we determine the effect of patatin-related phospholipase A pPLAIIIγ on *Arabidopsis* response to abiotic stress. Knockout of *pPLAIIIγ* rendered plants more sensitive whereas overexpression of *pPLAIIIγ* enhanced plant tolerance to NaCl and drought in seed germination and seedling growth. The *pPLAIIIγ-*knockout and overexpressing seedlings displayed a lower and higher level of lysolipids and free fatty acids than that of wild-type plants in response to NaCl stress, respectively. These results indicate that pPLAIIIγ acts a positive regulator of salt and osmatic stress tolerance in *Arabidopsis*.

## 1. Introduction

High salinity and drought are major and common stress conditions that limit crop production and threaten food security. Cellular membranes are initial sites of stress contacts and perception [1]. An increasing body of studies indicates that membrane lipid changes and enzymes involved play important roles in mediating plant response to salt and drought stresses [2,3]. Various phosphodiester-hydrolyzing phospholipases, including phospholipase D (PLD), phosphoinositide-specific phospholipase C (PI-PLC), and nonspecific PLC (NPC), are induced under stress and mediate plant response to salt and osmotic stresses [2,3,4,5,6,7,8,9,10].

Patatin-related phospholipase A (pPLA) is a major family of acyl-hydrolyzing enzymes that produce free fatty acids (FFAs) and corresponding lysoglycerolipids. Based on the gene structure and deduced protein sequences, thirteen patatin-related genes have been identified in *Arabidopsis* [11]. They are classified into four subfamilies such as pPLAI, pPLAII(α, β, γ, δ, ε), pPLAIII (α, β, γ, δ) and the fourth subgroup including SUGAR-DEPENDENT1 (SDP1), SDP1-L and adipose triglyceride lipase-like [11,12]. PLAI has only one member that contains a C-terminal leucine–rice repeat domain and an N-terminal ankyrin-like domain that are involved in protein–protein interaction. pPLAIIs contain five to six introns while pPLAIIIs possess only one intron. pPLAs have been implicated in many biologic processes, including lipid metabolism, signal transduction, cell growth and plant responses to environmental stresses [13,14,15,16,17,18]. pPLAI activity has been shown to increase under pathogen attack. Such rapid activation of pathogen-induced pPLA activity differentially affects biosynthesis of oxylipins [13,14,15]. pPLAIIα was found to be involved in drought stress [16], while functions of *pPLAIIγ, δ and ε* are shown in *Arabidopsis* root development in response to hormone and nutrient starvation [19].

pPLAIIIs contain a noncanonical esterase motif GxGxG instead of GxSxG in other family members [17,19]. The *pPLAIII* subfamily in *Arabidopsis* has four members, *pPLAIII(α, β, γ, δ).* Overexpression of *pPLAIIIβ* decreased cellulose content, anisotropic growth and plant height [17]. The decrease in longitudinal growth has also been found when *pPLAIIIδ* was overexpressed in *Arabidopsis*, camelina and canola [20,21]. Similarly, overexpression of *OsPLAIIIα* was found to decrease cell elongation in vegetative and reproductive growth in Rice [22]. In addition, knockout of *pPLAIIIδ* decreased whereas its overexpression increased seed oil content [18,21]. *pPLAIIIδ* deficiency in *Arabidopsis* impeded early auxin-induced gene expression and increases lateral root numbers under auxin treatment [23]. pPLAIII enzymes hydrolyze phospholipids and galactolipids to produce FFAs and lysolipids, such as lysophosphatidylcholine (LPC), lysophosphatidylethanolamine (LPE) and lysophosphatidylglycerol (LPG) [17,18]. The direct products of pPLAs are important cellular mediators. For example, LPC induced the activation of an H^+^/Na^+^ exchange transporter and the synthesis of phytoalexins [24]. FFAs promoted oxylipin production and inhibited plant root growth [17]. However, the role of *pPLAIIIγ* in plants is yet to be reported. In addition, despite the potential role of pPLAIIIs in plant growth and development, little is known about the effect of pPLAIIIs on plant response to abiotic stress conditions. Here we show that pPLAIIIγ plays a positive role in salt and drought stress.

## 2. Results

### 2.1. pPLAIIIγ-KO Seedlings Are More Sensitive to NaCl and Osmotic Stress

To determine *pPLAIIIγ* function, a homozygous T-DNA insertion mutant of *pPLAIIIγ* was identified (Figure 1a,b). qPCR analysis of seedlings detected a negligible level of the *pPLAIIIγ* transcript in the KO mutant (Figure 1c). The loss of expression in *pPLAIIIγ*-KO plants was also reported previously [18]. The KO seedlings grew and developed normally as did WT in the half-strength MS medium (Figure 1d). However, when seedlings were transferred to media containing 75 and 100-mM NaCl, primary root growth of KO seedlings was significantly more inhibited than that of WT (Figure 1d– f). *pPLAIIIγ-*KO seedlings also displayed more decrease in overall growth, as indicated by lower dry weights of total seedlings than WT under 75 and 100-mM NaCl (Figure 1e,f). The *pPLAIIIγ-*KO mutant was transformed with a construct containing pPLAIIIγ with its native promoter, the genetic complementation restored the KO mutant phenotype to that of WT (Figure 1d–f). The result indicates that the loss of *pPLAIIIγ* is responsible for increased sensitivity to NaCl stress.

In addition, *pPLAIIIγ-*KO seedlings are more sensitive to osmotic stress imposed by sorbitol and PEG (Figure 2). The primary root growth of *pPLAIIIγ-*KO seedlings was significantly more inhibited than that of WT in the media supplemented with 100, 150, 250 and 300-mM sorbitol and dry weight of total seedlings decreased significantly only at 150 mM treatment compared with that of WT (Figure 2b,c), suggesting that KO and WT seedlings displayed the most difference at 150-mM sorbitol than all the sorbitol concentrations tested. In addition, the primary root growth of *pPLAIIIγ-*KO seedlings was significantly more impaired than WT on the PEG-containing media with −0.40 and −0.7 MPa water potential (Figure 2a,d). Genetic complementation of *pPLAIIIγ-*KO restored the phenotype to WT (Figure 2e), indicating that the loss of *pPLAIIIγ* confers the increased sensitivity to osmotic stress.

### 2.2. Overexpression of pPLAIIIγ Enhances Tolerance to NaCl and Drought

To further determine the *pPLAIIIγ* function in abiotic stress, several *pPLAIIIγ*-OE lines were generated by placing *pPLAIIIγ* under the control of the cauliflower mosaic virus 35 s promoter. *pPLAIIIγ*-OE protein was tagged with GFP-His at the C terminus, and the production of pPLAIIIγ-GFP-His in plants was confirmed by immunoblotting using an anti-GFP antibody (Figure 3a). pPLAIIIγ-GFP was detected in leaf epidermal cells of *pPLAIIIγ*-OE lines, but not in WT (Figure 3b). To test whether pPLAIIIγ was associated with the cell wall, plasmolysis was induced in root cells with 0.5% NaCl. The GFP signal withdrew from the cell wall with protoplasts during plasmolysis, suggesting that it was not associated with cell wall (Figure 3c).

To examine the effect of pPLAIIIγ alterations on seedling growth in response to NaCl, 3 days seedlings grown on a half-strength MS medium were transferred to plates supplemented with 0-, 75- and 100-mM NaCl (Figure 3d,e). Ten days after transferring, primary root length and dry weight of KO plants decreased significantly compared to WT in 75- and 100-mM NaCl treatments (Figure 3f,g). However, under a normal growth condition, primary root growth and dry weight of overexpression lines were significantly inhibited in a half-strength MS medium (Figure 3d and Appendix A). To measure and calibrate the pPLAIIIγ effect on growth, the relative growth was calculated as data _treatment_/data _untreatment_ × 100%. At 75-mM NaCl treatment, root length was inhibited significantly by ~40% in KO, ~19% in WT and ~15% in OE lines (Figure 3f). When seedlings were treated with 100-mM NaCl, root growth decreased by ~58% in KO, ~38% in WT, ~30% in OE lines (Figure 3f). Similarly, dry weight displayed a tendency to reduce very significantly in KO seedlings, increase in OE seedlings compared with WT (Figure 3 g). The results indicate that compared with WT, the KO seedlings are more whereas OE seedlings are less sensitive to NaCl stress. When the seedling performance of WT, KO and OE lines in response to PEG stress was calculated, root growth was inhibited by ~62% in KO, ~32% in WT and ~20% in OE lines under the −0.40 MPa treatment (Figure 3f).

In addition, the effect of *pPLAIIIγ* on seed germination and seedling growth was tested by germinating WT, KO and OE seeds under different NaCl concentrations (Figure 4a). In the absence of salt stress, WT, KO and OE seeds displayed no difference in germination rates (Figure 4b). In the presence of 150 and 175-mM NaCl, the seed germination rate of KO mutant was lower whereas that of OE2 and OE3 lines was higher than WT at early stages of germination (Figure 4a,b). Compared with WT, KO seedlings exhibited a visible reduction of seedling growth whereas OE2 and OE3 seedlings displayed enhanced growth (Figure 4a 72, 96, 120 and 144 h). These data indicate that pPLAIIIγ plays a positive role in abiotic stress tolerance.

### 2.3. pPLAIIIγ Knockout Plants Are Hypersensitive to NaCl and Drought in Soil

To further test the pPLAIIIγ function in stress responses, WT, KO and OE plants were grown side by side under NaCl and drought stress, as well as no stress control, in soil to evaluate plant growth and survived rate (Figure 5a). Without applied stress, the fresh weight of KO plants was higher whereas that of OE lines was lower significantly than that of WT (Appendix A). In contrast, after NaCl treatments for 20 days, the fresh weight of KO plants was lower than that of WT (Appendix A). When the relative growth was calculated, the fresh weight of KO plants was lower significantly than WT whereas OE plants displayed a smaller decrease compared with WT, suggesting that pPLAIIIγ-OE plants was more tolerant to NaCl (Figure 5b).

WT, KO and OE plants exhibited wilted leaves when three-week-old plants were deprived of water for 16 days (Figure 5a,c). KO plants accumulated more anthocyanin under this condition (Figure 5a). These drought-treated plants were rewatered and scored for the rate of survival 18 days after rewatering. Whereas most of wilted KO plants failed to recover, more than 50% of wilted WT and OE survived, with a lower survival rate in KO than WT plants (Figure 5c). These data indicate that *pPLAIIIγ* positively affects *Arabidopsis* tolerance to salt and drought in soil.

### 2.4. Loss of pPLAIIIγ Decreases Lysolipids and Free Fatty Acid Contents

To investigate the effect of pPLAIIIγ on lipid metabolism under salt stress, lipid content and molecular species were determined using electrospray ionization tandem mass spectrometry (ESI-MS/MS). Total lipids were extracted from 15-day-old seedlings grown in plates treated with 0- or 100-mM NaCl for 0 or 3 h. Under normal growth conditions, total LPC content was significantly lower in KO and higher in OE lines compared with WT (Figure 6a, left panel). After salt treatments, total LPC content of WT, KO and OE lines slightly increased compared with the non-salt treatment, the LPC content was still lower in KO and higher in OE lines than in WT (Figure 6a, left panel). LPE exhibited a similar tendency as did LPC, being decreased in KO and increased in OE compared with WT (Figure 6a, right panel). Without NaCl, the content of LPCs with 16:1, 16:0, 18:1 and 18:0 acyl species in KO mutants was significantly lower than that of WT (Figure 6b). The level of LPEs with 16:0 and 18:1 in KO was lower, whereas that of LPEs with 16:1, 18:3 and 18:2 in OE lines was higher than WT (Figure 6b). After the NaCl treatment, the content of LPCs with 16:0, 16:3 and 18:1 in KO decreased significantly, whereas that of LPCs with 16:0, 16:1, 16:3 and 18:2 in OE lines was higher significantly than WT (Figure 6b). The level of LPE species in KO was lower but tended to be higher in OE lines compared with that of WT (Figure 6b).

Under the normal condition, FFA content in KO seedlings was lower whereas that in OE lines was higher than that in WT (Figure 6c). After the NaCl treatment, an increase of total FFA in WT, KO and OE lines was observed compared with control (Figure 6c). The total FFA content was ~25% lower in KO mutant and 14% higher in OE lines than that of WT (Figure 6c). All FFA species tended to decrease in KO mutant and increase in OE lines compared with WT (Figure 6d). These data indicate that *pPLAIIIγ* contributes to the production of lysolipids and FFAs in seedlings, particularly in response to NaCl stress.

### 2.5. Expression of pPLAIIIγ in Response to Salt and Its Effect on the Expression of Salt Response-Related Genes

To monitor the effect of *pPLAIIIγ* alterations on the expression of genes involved in salt responses, WT, KO and OE plants were grown hydroponically for 4 weeks, and then treated with 0 and 100-mM NaCl. Leaves and roots were collected from 0, 1, 3, 6, 12 and 24 hours after treatments. In roots, the level of *pPLAIIIγ* transcript increased 1 or 3 h after NaCl treatment whereas in leaves, only an increase occurred after 24 h of treatments (Figure 7). The transcript level of *pPLAIIIγ* was higher in roots than in leaves in early time points, consistent with microarray data in Genevestigator (http://www.genevestigator.com).

The expression of genes involved in the salt overly sensitive (SOS) signal transduction pathway genes was assayed as this pathway is probably the best characterized in mediating salt-specific responses [25,26]. In WT roots, the level of *SOS2* transcript increased 1 h after the NaCl treatment whereas that of *SOS3* and *CBL10* transcripts increased 3 h after the salt treatment (Figure 7). However, in KO roots, the levels of *SOS2*, *SOS3* and *AtCBL10* transcripts decreased significantly at some timepoints compared with WT (Figure 7). The transcript level of *SOSs* increased sharply at some time points of NaCl treatment in OE roots (Figure 7).

## 3. Discussion

The effects of ablation and overexpression suggest that pPLAIIIγ plays a positive role in *Arabidopsis* response to salt and osmatic stress. Compared with WT, *pPLAIIIγ*-knockout plants were more sensitive to NaCl and drought, while OE lines displayed the opposite effect to that of knockout on seed germination, seedling growth in plates and plant growth in soil. This raises many intriguing questions regarding how pPLAIII affects plant function. Plant PLAs are induced rapidly by different signals to produce lysolipids and FFA functioning as second messengers that regulate distinct proteins or downstream process and are thought to be important for early signal events [27]. Calcium-dependent protein kinases was reported to be an upstream signal for pPLA activation, for example, pPLAIIε and pPLAIIδ were activated by specific phosphorylation in their C-terminal sites by CPK3 and CPK4, respectively [19]. In addition, Ca^2+^ ion influx channel and tobacco phospholipase D (PLD) are thought to be upstream signals of pPLAs in response to biotic or abiotic stresses [28]. The activity of pPLAs is partially regulated by pH [29]. To date, little is known specifically about upstream signals of pPLAIIIs in response to biotic or abiotic stresses.

The direct products of pPLAs are lysolipids and FFA. Lysolipids such as LPCs are highly mobile within intact cells, and therefore considered to be a good candidate for a cytoplasmic messenger that transduces signals to activate downstream processes and gene expression in the nucleus [30]. The previous studies suggest that the downstream targets of lysolipids could be specific receptors, protein kinases, protein phosphatases or other signaling enzymes. For example, PLDα was found to be specifically inhibited by lysolipids generated by PLA_2_, resulting in plant senescence [31,32]. H^+^-ATPases was reported as potential receptors for lysophospholipids in plants [33]. LPC as signal molecule was further demonstrated during mycorrhiza initiation in potato [30]. This study showed that *pPLAIIIγ*-OE seedlings had shorter primary roots and seedling height and higher levels of FFAs and lysolipids than did WT. The elevated intracellular level of FFAs may, at least in part, result in the dwarf phenotype. This result is consistent with the previous studies that FFA inhibited plant growth [17]. Lysophospholipids have also been reported to inhibit cell and root elongation [17]. Therefore, a combination of higher levels of FFAs and lysophospholipids may lead to decreasing primary root growth and seedling height. However, the downstream targets of lysolipids and FFAs generated by pPLAIIIs remain to be identified.

One potential target of the pPLAIIIγ-derived lysolipids and FFA could be the mitogen-activated protein kinases 6, MPK6 [34] and the activated MPK6 phosphorylate the Na^+^⁄H^+^ antiporter, SOS1, which may contribute to reduce Na^+^ accumulation in plants [34]. In *pPLAIIIγ* knockout mutant, the relative expression levels of SOS2, SOS3 and AtCBL10 were diminished under salt treatment compared with that of WT. In contrast, the level of *SOSs* and *AtCBL10* in OE lines were increased at some time points of salt treatment. It would be of interest to examine whether pPLAIIIγ alters abiotic responses via modulating MAPK, SOS and other cascades. In addition, OE lines displayed higher levels of unsaturated 18 carbon fatty acids than did WT. These FFAs, such as C18:3, can be substrates for JA and other oxylipin production [17]. JA is known to inhibit root growth and JA signaling is activated in the roots at early stages of the salt stress response [35]. This raises further question that FFAs increased tolerance to abiotic stress due to the conversion from FFAs to JA. Further future studies are needed to understand upstream activators and downstream effectors mediated by pPLAIII and its products.

## 4. Materials and Methods

### 4.1. Generation of Knockout, Overexpression and Complementation Plants

Seeds of a T-DNA insertional mutant of *pPLAIIIγ* (Salk_088404; http://signal.salk.edu/cgi-bin/tdnaexpress) were obtained from the *Arabidopsis* Biologic Resource Center (ABRC; Ohio State University). The homozygous T-DNA insertion mutant was confirmed by PCR using a mix of three primers: LP, BP and RP (Appendix A), plants that were confirmed as homozygous T-DNA insertion mutant were further verified by qPCR using primers JLP053 and JLP054 (Appendix A). To generate the *pPLAIIIγ* overexpression construct, the genomic DNA spanning the coding region was PCR-amplified using Col-0 *Arabidopsis* genomic DNA as a template and the primers JLP023 and JLP024 (Appendix A). The product was cloned into the pMDC83 vector before the GFP-His coding sequence (GFP-6x histidine). The construct was introduced into wild-type *Arabidopsis* (Col-1) plants using agrobacterium-mediated transformation. Positive transformants were identified initially by selection on 1/2 MS media with hygromycin and confirmed by PCR. To genetically complement the *pPLAIIIγ* KO, the *pPLAIIIγ* genomic region with or containing its native promoter (−1537 bp from the start codon) was PCR-amplified using primers JLP027 and JLP028 (Appendix A) and cloned into the binary vector pEC291. Positive transformant seedlings were identified initially by selection on 1/2 MS media with hygromycin and confirmed by PCR.

### 4.2. Plant Growth Conditions and Treatments

Seeds of *Arabidopsis thaliana* Columbia-0 (WT), *pPLDIIIγ* knockout (KO) and overexpression (OE) lines were sterilized using 75% ethanol for 10 min and then 20% bleach for 10 min and rinsed four times with sterile distilled water. Sterilized seeds were sowed on plates and stratified for 3 days in dark at 4 °C, and then the plates were moved to a growth chamber with a 16-h-light/8-h-dark cycle and at 23/21 °C, under cool fluorescent white light (200 μmol m^−2^ s^−1^). Three days later, seedlings were transferred to plates and grown vertically in the same growth chamber. For NaCl treatments, plants were grown on agar plates on half-strength Murashige & Skoog (MS) media containing 2.2-g L^−1^ MS basal salts, 1% (*w/v*) sucrose, 1% (*w/v*) agar supplemented with 0-, 75-, 100-, 150- and 175-mM NaCl. For sorbitol treatments, sucrose content in media was reduced to 0.5% with 0-, 100-, 150-, 250- and 300-mM sorbitol. Plates with −0.25- (control), −0.40- and −0.70-MPa polyethylene glycol (PEG) were prepared according to Peters, et al [6]. Briefly, half-strength MS media containing 1.5% agar, 0.5% sucrose were solidified and then overlaid with a solution containing 6 mM MES with 0-, 170-, 400-g L^−1^ PEG (molecular weight: 8000, Sigma-Aldrich), which gave rise to −0.25 (control), −0.40 and −0.70 MPa, respectively. After the PEG solution was left on plates for 24 h, excessive PEG solution was removed from plates. Control plates were made with half-strength MS media containing 1% agar and 0.5% sucrose. Three biologic repeats of each treatment were analyzed. Primary root length was measured with ImageJ software (v1.44; National Institutes of Health, Bethesda, MD, USA). Seedling were harvested after taking images and then oven-dried at 100 °C for 2 days. Dry weight of 20 seedlings was weighed as a biologic repeat of three biologic replicates. In order to measure relative growth, data (primary root length or fresh/dry weight) were calculated based on the following formula, relative growth = data _treatment_/data _control_ × 100%.

### 4.3. Hydroponic Cultivation

Seeds were stratified as described above and sowed on sponge pots. Plants were cultured in 72-plug trays in modified half-strength Hoagland’s solution [36]. Three liters of solution were replaced for the first time in one week, and thereafter twice per week. The plants were grown at 23 °C under with a 16-h-light/8-h-dark cycle as described as above. After 4 weeks sowing on trays, the plants of WT, KO and OE line mutants were treated with Hoagland solution supplemented with 0, 100-mM NaCl. Leaves and roots were harvested for RNA extraction after treatments.

### 4.4. Salt and Drought in Soil

Seeds of WT, KO and OE-lines were stratified in 4 °C for 3 days and germinated in soil (PRO-MIX FPX). Three-week-old plants were watered with 0- or 100-mM NaCl twice per week. After 20 days, plants were harvested to measure fresh weight. The relative growth was calculated based on the formula as shown above. For drought treatments, two-week-old plants were withheld of watering for 16 days and the number of wilted plants was recorded. Then these drought-treated plants were watered and after 18 days of rewatering, the number of survival plants was scored. Plants were grown in a growth camber under the same conditions as described above.

### 4.5. Germination Test

*Arabidopsis* seeds were sowed on agar plates containing 1/2 MS medium supplemented with 0-, 150- and 175-mM NaCl. Thirty-seven to forty-one seeds per WT-, KO- and OE-line were sowed in one plate, three biologic repeats were performed in germination test. The growth conditions were the same as that for seedling growth in growth chamber. Germinated seeds, the visible sprouting of a seedling from seed, were counted on 24, 48, 72, 96, 120 and 144 h after stratification.

### 4.6. RNA Extraction and Quantitative Real time (qPT)-PCR

RNA extraction and qPCR were performed according to Li et al. [17]. Briefly, hydroponically grown 4-week-old plants were used for expression analysis. For the determination of the basal expression of the genes, root and leaf samples were collected immediately prior to the exchange nutrient solution. The original solution was replaced with fresh solution with 0 and 100 mM NaCl. The samples were harvested on 1, 3, 6, 12 and 24 h after changing fresh solution and instantly frozen into liquid nitrogen. RNA was isolated using TRIzol reagent (Life Technologies) according to the manufacturer’s instructions. DNA contamination in RNA samples was removed with RNase-free DNase (Ambion, Inc. Austin, TX, USA). RNA was reverse-transcribed using iScript cDNA Synthesis Kit (Bio-Rad, Hercules, CA, USA). Quantitative RT-PCR (qPCR) was performed with MyiQ Sequence Detection System (Bio-Rad) using SYBR Green to monitor the expression of the target genes and reference gene. UBQ10 was used as reference gene for the normalization of target gene expression. Fold change in expression of the target gene was calculated using (1 + E)^∆(∆Cp)^ method [37] Primers are described in Appendix A.

### 4.7. Lipid Profiling

Lipids were extracted and quantified according the methods [38]. Briefly, the seedlings of WT, KO and OE lines were grown vertically in plates with half-strength MS medium for 15 days. Then, the seedlings were treated with NaCl by immersing roots in 1/2 MS nutrient solutions supplemented with 0- or 100-mM NaCl for 0 or 3 h, respectively. The seedlings were collected and immersed immediately into 3 mL of hot isopropanol containing 0.01 (*v/v*) butylated hydroxytoluene at 75 °C for 15 min. Lipids were extracted five times from samples using chloroform–methanol (2:1 *v/v*), the remaining tissues were oven-dried at 100 °C for two days and weighed. Lipid composition was analyzed by ESI-MS/MS (API4000; AB SCIEX, Foster City, CA, USA). Plant lipid extract in chloroform was combined with solvents, chloroform-methanol-ammonium acetate (300 mM) in water (300:665:35 *v/v*) and internal standards. Lipids contents were calculated by comparison of standards and amounts of lipids were normalized based on the delipidated dry weight of samples [39]. Free fatty acids (FFAs) were determined by ESI-MS, using the deuterated internal standard (7,7,8,8,-d4-17:0 fatty acid) (Sigma-Aldrich) by scanning in the negative ion mode in a mass range of m/z 200 to m/z 350 [17].

## Figures and Tables

**Figure 1 plants-09-00650-f001:**
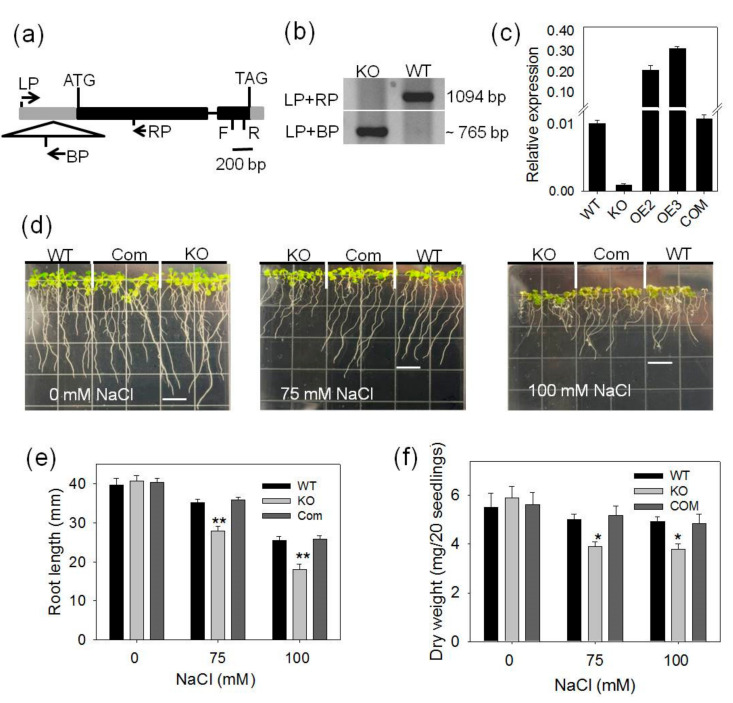
*pPLAIIIγ*-KO, its complementation (COM) and response to different NaCl concentrations; (**a**) T-DNA insertion site in *pPLAIIIγ*. Solid boxes indicate exons, thin lines indicate introns, gray boxes represent the untranslated regions (UTR) at both the 5′ and 3′ positions. Arrows (LP, BP and RP) show positions of primers used for PCR. F and R present positions of JLP053 and JLP054 primers, respectively, for *pPLAIIIγ* relative expression in qPCR; (**b**) PCR-confirmation of T-DNA insertion in *pPLAIIIγ* using genomic DNA; (**c**) relative gene expression of *pPLAIIIγ* in WT, KO, OE and COM lines by qPCR. *UBQ10* was used as internal control. Data are means ± SE (*n* = 3); (**d**) representative images of seedlings grown on half-strength MS media supplemented with 0, 75 and 100-mM NaCl. Measurement of primary root length (**e**) and dry weight (**f**) of WT, KO and COM lines. Seedlings were grown for 7 days under different treatments after transferring and then measured. Values are mean ± SE (n = 45–60 for root length or 3 biologic repeats for dry weight. One and two asterisks indicate significant differences from WT by Student’s t-test at *p* = 0.05 and *p* = 0.01, respectively.

**Figure 2 plants-09-00650-f002:**
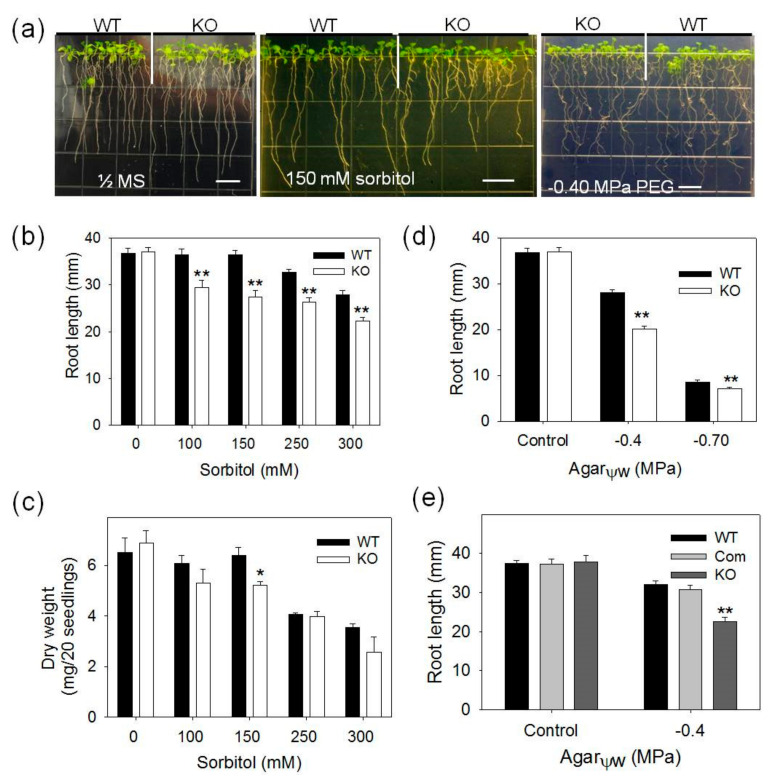
Responses of *pPLAIIIγ*-KO, COM and WT *Arabidopsis* plants to sorbitol and polyethylene glycol (PEG). (**a**) Representative images under different sorbitol and PEG concentrations; (**b**, **c**) quantification of seedling growth in response to different levels of sorbitol. Three-day-old seedlings germinated on ½ MS media after stratification were transferred to plates with 0, 100, 150, 250 and 300-mM sorbitol. Seedlings were harvested and measured 7 days after transferring; (**d**,**e**) quantification of seedling growth in response to different levels of PEG. Values are mean ± SE from three biologic repeats for dry weight of each line in (**c**). Error bars indicate the SE (*n* = 45–60) in (**b**,**d**,**e**) for primary root length. One and two asterisks indicate significant differences from WT by Student’s t-test at *p* = 0.05 and *p* = 0.01, respectively.

**Figure 3 plants-09-00650-f003:**
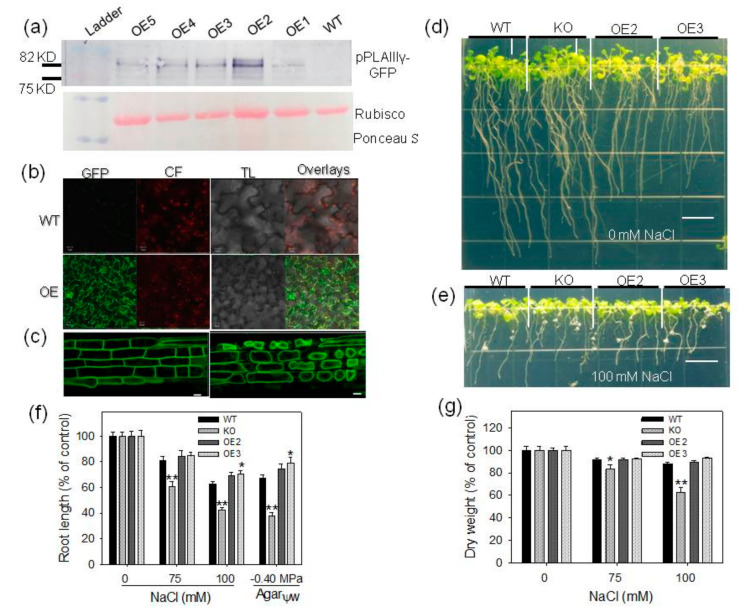
Opposite effects of *pPLAIIIγ-*KO and -OE lines on tolerance to NaCl and PEG; (**a**) Immunoblotting of GFP-His-tagged pPLAIIIγ in *Arabidopsis*. Five independent homozygous lines of *pPLAIIIγ*-OE mutants were examined; (**b**) confocal images of epidermal cells of WT and *pPLAIIIγ*-OE:GFP leaves and chlorophyll fluorescence versus GFP; (**c**) plasmolysis of root epidermal cell of the *pPLAIIIγ*-OE:GFP. Representative images of WT, KO and OE lines without (**d**) or with (**e**) NaCl stress; (**f**) relative growth of root length of WT, KO and OE lines. Values are mean ± SE (*n* = 45–60); (**g**) relative growth of dry weight of WT, KO and OE lines. Values are mean ± SE from three biologic repeats for each line (20 seedlings dry weight per biologic repeat). One and two asterisks indicate significant differences from WT by Student’s t-test at *p* = 0.05 and *p* = 0.01, respectively.

**Figure 4 plants-09-00650-f004:**
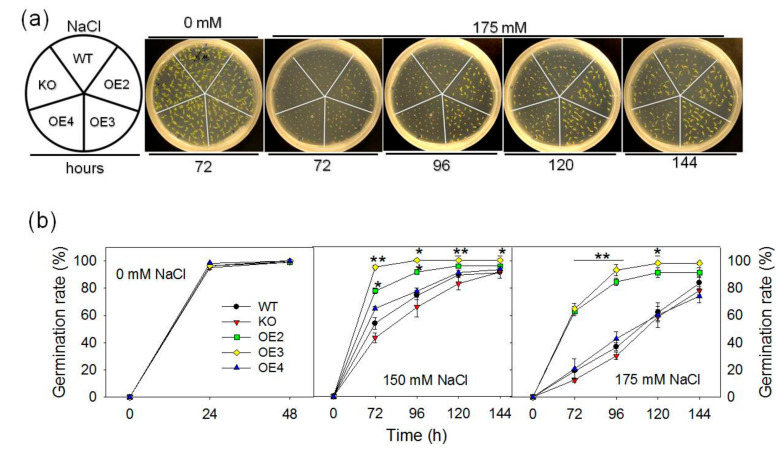
Effect of *pPLAIIIγ*-KO and -OE on seed germination under salt stress. (**a**) Representative images of seed germination; (**b**) effect of NaCl on seed germination phenotype of wild type (WT), *pPLAIIIγ* knockout (KO) and OE-lines. Thirty-seven to forty-one seeds of each line per biologic repeat were germinated on an agar plate containing 0-, 150- or 175-mM NaCl, respectively. Germinated seeds were counted at 48, 72, 96,120 and 144 h after stratification. Values are mean ± SE of three biologic repeats. One and two asterisks indicate significant differences from WT by Student’s t-test at *p* = 0.05 and *p* = 0.01, respectively.

**Figure 5 plants-09-00650-f005:**
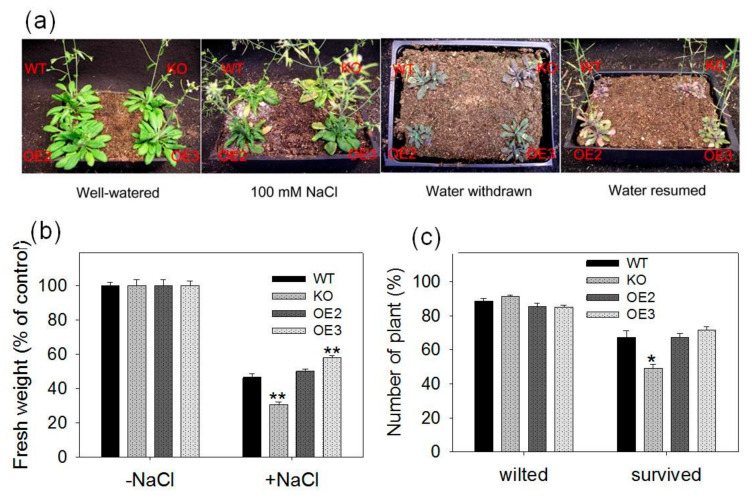
Soil-grown plants of WT, *pPLAIIIγ*-KO and OE mutants under salt and drought stress; (**a**) images of plants that were watered regularly *(well*-*watered),* treated with 100-mM NaCl for 20 days (100-mM NaCl) or deprived of water for 16 days *(water withdraw)* and then watering was resumed *(water resumed).* Three-week-old seedlings were used for NaCl and drought treatments, and drought-treated plants were rewatered and grown for 18 days; (**b**) measurement of the relative growth of WT, KO and OE lines; (**c**) measurement of wilted and survived plants when water was withdrawn (*wilted*) and survived plants that wilted plants were watered again when water was resumed (*survived*). Error bars represent standard error (four biologic repeats). One and two asterisks indicate significant differences from WT by Student’s t-test at *p* = 0.05 and *p* = 0.01, respectively.

**Figure 6 plants-09-00650-f006:**
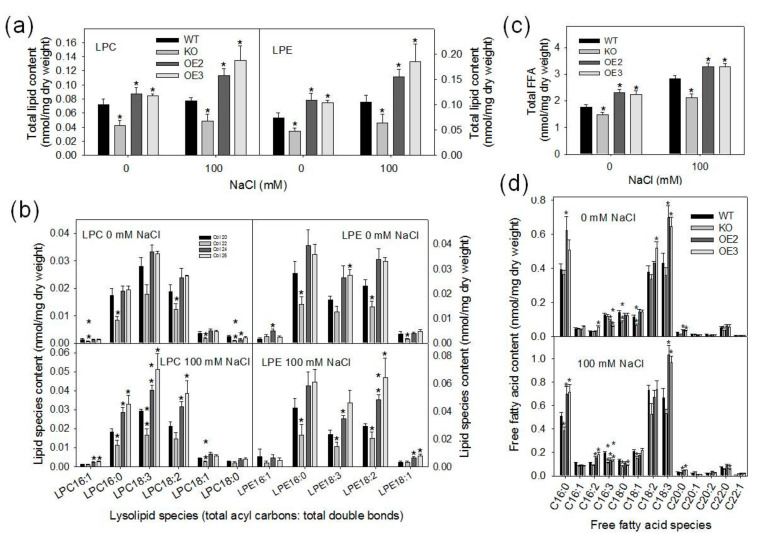
Effects of *pPLAIIIγ*-KO and OE on lysophospholipid and free fatty acids (FFA) levels in *Arabidopsis*; (**a**) total lysophospholipid content of LPC and LPE in WT, *pPLAIIIγ*-KO and -OE seedlings; (**b**) lysophospholipid species in WT, KO and OE lines; (**c**) total FFA content in WT, KO and OE lines; (**d**) FFA molecules species in WT, KO and OE lines. Values are means ± SE (*n* = 5 separate samples). Asterisks indicate significant differences from WT by Student’s t-test at *p* = 0.05.

**Figure 7 plants-09-00650-f007:**
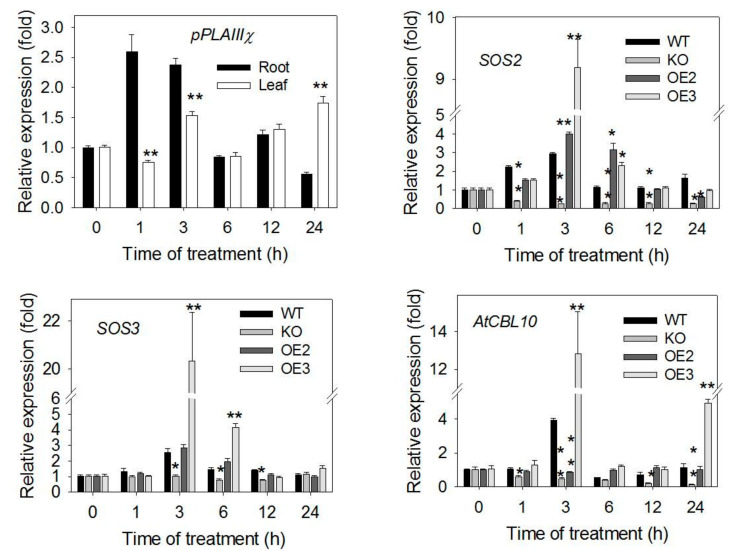
Transcript levels of *pPLAIIIγ* and SOS genes in WT, KO and OE lines under 0 and 100-mM NaCl. Transcript level of *pPLAIIIγ* measured by qPCR in leaves and roots of WT with and without 100-mM NaCl at different time points (1–24 h). Transcript levels of *SOS2*, *SOS3*, *AtCBL10* quantified by qPCR in roots of WT and KO seedlings with treatments as above. *UBQ10* was used as internal control and data represent means ± SE (*n* = 4 technical repeats). asterisk presents a significant difference at 0.05 level by Students’ t-test.

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
