# Peer review of "Patatin-Related Phospholipase pPLAIIIγ Involved in Osmotic and Salt Tolerance in Arabidopsis"

_plants, 2020, doi:10.3390/plants9050650_

Round 1

Reviewer 1 Report

Please see my comments and suggestions in the attached file.

Reviewer 2 Report

Comments to the author

General comments:

1)      The terms ‘osmotic tolerance’ or ‘osmotic stress’ are misleading. In your study, you have analyzed the response to osmotic relevant substances like sorbitol and PEG but also to NaCl. The response to salt does not contain solely an osmotic component but also the salt response per se. The term abiotic stress tolerance appears more suitable.

  • The presentation of your results is not stringent and therefore confusing. After presenting stress responses of the different pPLAIIIγ genotypes on agar plates and during seed germination, you show the effect of pPLAIIIg knock-out and overexpression on acyl-hydrolase products lysolipids and free fatty acids. Subsequently, again stress response results like expression profiles of salt stress response genes in the different pPLAIIIg genotypes and the stress response of soil-grown plants are presented. Neither ‘Results’ nor ‘Discussion’ give explanations for this order of results.
  • All images are too small, especially the confocal images shown in figure 3 b and c.
  • The discussion of the results you obtained has to be intensified:
    1. The PLAIIIg overexpressing genotypes show reduced root growth under control conditions. Accordingly, you are calculating relative growth. Please, discuss potential reasons for this altered phenotype and impacts of this phenomenon on your study.
    2. You measured the content of the acyl-hydrolase products lysolipids and free fatty acids but not the reaction substrates. Please, discuss potential substrates and why you did not measure them.
    3. The content of lysolipids and free fatty acids is altered in the PLAIIIg knock-out and overexpressor genotypes. Please, discuss if this means that PLAIIIg is a functional acyl-hydrolase and if these findings are in line with previous findings by other authors.
    4. Why did you measure the content of lysolipids and free fatty acids and the specific molecular species? How does this help to explain the function of pPLAIIIg in response to abiotic stress?
    5. Transcript levels of SOS-pathway components are altered in the various pPLAIIIg genotypes indicating a transcriptional regulation and an interplay. Are there other mechanisms possible how pPLAIIIg modifies salt/abiotic stress response?

Abstract

L12: Please specify what you mean by “various physiological processes”.

L15: Please use the term osmotic stress instead of drought as you applied osmotic stress by the addition of substances like sorbitol and PEG.

L16-18: The content of lysolipids and free fatty acids is altered under control and salt stress conditions. Accordingly, the results indicate that pPLAIIIg acts as a acyl-hydrolase and a positive regulator of abiotic stress tolerance.

 Introduction

L29: …are induced under …

L36: The PLAI family consists of only one member that …

L40-42: pPLAI (PLP2) activity has been shown to increase under pathogen attack. Such rapid activation of pathogen-induced pPLA activity …

Results

L59: Please, use osmotic stress instead of drought stress.

L60: T-DNA insertion

L64: Plant growth on 50 mM NaCl is not shown in figure 1 d and e.

L67-69: Please, rephrase the sentence: The pPLAIIIg-KO mutant was transformed with a construct containing pPLAIIIg…

Fig. 1: All letters (a, b, c, d) should be distributed equally at their corresponding figures. Accordingly, c should be written on top of the graph. L73: Please, mention the exact T-DNA insertion site and that this site is demonstrated by the triangle.

L78: grown on half-strength…L81-82: Students’ t-test

L84-85: … significantly more inhibited…

L86: Is there an explanation, why the alterations of plant dry weight were dissimilar in response to sorbitol treatment?

Fig. 2b: Please, check if the upper and the lower image have the same size.

L95 and L98: Quantification or determination instead of qualification

L107-108: Which concentration of NaCl was used to induce plasmolysis?

L112: 10 days after transferring, primary root length …

L113: … decreased significantly compared to WT…

L119-120: Similarly, dry weight displayed a tendency…

Fig 3.: Numbering of figures 3 f and g is missing.

L126: Response to sorbitol is not shown in that figure.

L129: …and versus GFP?

L132: Relative growth rate was already defined in ‘Material and Methods’ and ‘Results’. The repeated definition is not required here.

L139: Why did you choose here NaCl concentration for the germination assay than for the seedling growth assay?

L143: Seed germination assays were conducted on NaCl. So, seeds are not only exposed to osmotic but also to salt stress.

L147: Thirty-three or four…. Please specifiy.

L148: agar plates containing 0, 150 or 175 mM NaCl, respectively.

L149: Please, give information on the statistics used here. What do the asterisks mean?

L150: Please, introduce the abbreviation for free fatty acids (FFA) here.

L154: Under normal growth conditions with NaCl…. Please, specify. Did you mean control conditions?

L171: Please, give information on the statistics used here. What do the asterisks mean?

L180: Please, specify which gene expression was monitored.

L182: It is confusing to readers, if you mention the roots first in the text but in the graph transcript levels in the leaves are shown first. Transcript levels increased after 1 and 3 h or 3 and 24 h respectively!

L186-188: SOS pathway is well-characterized. Is there link of this pathway to pPLAIIIs?

Fig6: Statistics are lacking in this figure. Are there significant differences?

L197: quantified instead of qualifiedL200: abiotic instead of osmotic

L202: survival instead of survived

L211: …watering was resumed

L215: Student’s t-testL218: Put the statement on the statistics used at the end of the paragraph if statistics were similar for all graphs shown.

L219-220: There are hardly any differences visible in the graph! Please, explain your statement.

Fig 7.: Are there significant differences in figure 7c?

Discussion

see general comments

L227: abiotic instead of osmotic; …plants were more sensitive…

L232-240: Here, you summarize potential upstream signals of PLAIIIs found by other authors. How is this linked to the function of PLAIIIg in plant stress response? This paragraph better fits as an outlook on future studies.

Materials and Methods 

L266: spanning instead of spinning

L270 and L274: … positive transformants…

L272: with or containing

L280: Please, state the conditions, under which the plants were grown in the growth chamber during the first 3 days.

L283: 4.4 g L-1 is the concentration for full-strength Murashige and Skoog medium

L284: Please check the NaCl concentrations you used in your study. Results of plants grown on 25, 50 and 200mM NaCl are not shown!

L288-289: Please check the PEG concentrations you used in the study! How much PEG-MES solution did you apply to the different plates.

L293-294: Please state clearly, how many replicates you did and how many independent repeats per experiment you performed.

L306: How much NaCl-solution was applied to the plants?

L314: Does line mean genotype?

L324: Please, give a reference for the cetyl-trimethyl-ammonium bromide method.

L332-333: Were plants transferred to new agar plates or treated hydroponically? 

Supplementary material 

L342-345: Please, check this paragraph for consistency. Figure S1b is not mentioned in the text

Reviewer 3 Report

This manuscript investigates the role of phospholipase pPLAIIIγ in molecular and physiological responses to osmotic stress in Arabidopsis thaliana. The authors used the corresponding null mutant and over-expressing lines to demonstrate that pPLAIIIγ reduces the effect of salt and osmotic stress on primary root growth and seed germination, and enhances the salt-induced transcription of a particular subset of responsive genes. They also verified that pPLAIIIγ facilitates the production of lysolipids and free fatty acids. The experiments were carried out appropriately and were replicated several times. The conclusions were based on statistical evaluation of the data obtained. The paper is technically sound, I have only a few comments on a missing control and the way of data presentation, in the order of their appearance in the text.

  1. Figure 1a: please indicate the app. position of qPCR primers (JLP053 and JLP054) on this drawing.
  2. Figure 2a: a loading control (eg. actin, tubulin, or other housekeeping protein) must be shown along the anti-GFP-specific bands.
  3. Figure 4b: please use colour graphs so the different lines could be distinguished easier.
  4. Section 2.3. It is clear that pPLAIIIγ affects lipid levels in normal (not salt-stressed) conditions. On the other hand, this section is concluded by this sentence: “These data indicate that pPLAIIIγ contributes to the production of lysolipids and FFAs in seedlings, particularly in response to NaCl stress.” This suggests that pPLAIIIγ also affects the effect of salt stress on lipid levels. In other words: the relative change in the level of a particular lipid species caused by salt is different in the WT, KO and OE lines. This can be tested if relative data for Figure 5b and 5d are shown ([lipid levels at 0 mM NaCl]: [lipid levels at 100mM NaCl]*100), even as a Supplementary Figure.
  5. Line 189: I cannot see the increase of SOS1 mRNA in the WT in response to salt.
  6. Figure 7c. The Y-axis label says: “Number of plants (%)”. However, the values look like ratios of the total and wilted or survived plants. Either modify the label or multiply values by 100.
  7. The results presented here may indicate that the elevated level of some lipid species could contribute to the management of salt stress. I wonder if there are published data on the role of lysolipids and FFAs on the osmotic stress response. because those could link pPLAIIIγ to the actual stress response. This could fit in the Discussion, which is quite short in the present version.

Round 2

Reviewer 1 Report

The manuscript was greatly improved and I am satisfied with the responses from the authors.

I just would like to mention that the title and legend of the Figure 5 need to be completed in the latest pdf.

Author Response

Please see the attachment, thanks!

Reviewer 3 Report

The authors adequately addressed my concerns, provided novel data, and modified the text as requested. I have no further comments.

Author Response

Thank you so much!